# Citrus Varieties with Different Tolerance Grades to Tristeza Virus Show Dissimilar Volatile Terpene Profiles

**Salvatore Guarino** [1], **Loredana Abbate** [1], **Francesco Mercati** [1,*], **Sergio Fatta Del Bosco** [1], **Antonio Motisi** [1], **Mokhtar Abdulsattar Arif** [2], **Gabriele Cencetti** [3], **Eleonora Palagano** [3] and **Marco Michelozzi** [3]

1   Institute of Biosciences and Bioresources (IBBR), National Research Council of Italy (CNR), Corso Calatafimi 414, 90129 Palermo, Italy; salvatore.guarino@ibbr.cnr.it (S.G.); loredana.abbate@ibbr.cnr.it (L.A.); sergio.fatta@ibbr.cnr.it (S.F.D.B.); antonio.motisi@ibbr.cnr.it (A.M.)
2   Plant Protection Directorate, Ministry of Agriculture, Abu-Ghraib, Baghdad 10081, Iraq; mokhtar.a.arif@gmail.com
3   Institute of Biosciences and Bioresources (IBBR), National Research Council of Italy (CNR), Via Madonna del Piano 10, 50019 Sesto Fiorentino, Italy; gabriele.cencetti@cnr.it (G.C.); eleonora.palagano@ibbr.cnr.it (E.P.); marco.michelozzi@cnr.it (M.M.)
*   Correspondence: francesco.mercati@ibbr.cnr.it

**Abstract:** Plants produce considerable amounts of volatile organic compounds (VOCs) with several biological functions, including protection against biotic agents such as viruses and their vectors. In citrus species, these metabolites can be related with their different susceptibility/tolerance toward the Tristeza virus (CTV), one of the main biotic constraints for the citrus industry. The objective of this study was to compare the VOCs pattern from the leaves of a CTV-susceptible citrus variety such as *Citrus aurantium* and from three CTV-tolerant varieties: *Citrus volkameriana*, Carrizo citrange, and Forner-Alcaide no. 5. The VOCs emitted were analyzed via the headspace SPME method, while plant metabolites sequestered in the leaves were analyzed by heptane extraction followed by GC-MS. The results indicated that the majority of the VOCs emitted and sequestered in the leaves of the varieties tolerant and susceptible to CTV are constituted mainly by volatile terpenes (VTs) that exhibit strong qualitative/quantitative differences among the profiles of the four citrus species. In detail, the VOC emission indicated different patterns between *C. aurantium* and *C. volkameriana* and from both of them in comparison with Forner-Alcaide no. 5 and Carrizo citrange that exhibited more similarities, with the last two characterized by a higher presence of sesquiterpenes. The data obtained from the analysis of the VOCs sequestered in leaf tissues of the CTV-tolerant varieties indicated a higher presence of monoterpenes such as limonene, α-pinene, and p-cymene, known to be the main components of several plant extracts showing deterrent properties toward viruses and insect vectors. As VOC evaluation is a fast and noninvasive measure of phenotypic dynamics, allowing the association of plant phenotypes in accordance to plant disease resistance and/or stress tolerance, the possible implications of such differences in terms of tolerance grade to CTV and/or its related vectors are discussed.

**Keywords:** plant secondary metabolites; VOCs; rootstock; CTV; *Citrus aurantium*; *Citrus volkameriana*; carrizo citrange; Forner-Alcaide no. 5

## 1. Introduction

Plants produce and emit a large variety of secondary metabolites with several functions, including communication and protection against biotic agents [1,2]. These compounds are known to have a primary role in the ecological interactions between plants and their phytophagous insects [3,4]. In particular, plants release a broad spectrum of volatile organic compounds (VOCs) to communicate with each other and to attract or repel insects, allowing "plant–plant" and "plant–insect" communities communications [1]. The VOCs

blend released strongly influence herbivore insects' behavior as pests rely heavily on olfaction for sensing their external environment [5]. Indeed, VOC profile modulations implicate changes in the plant defense response to herbivore feeding [6,7]. Insects exploit plant VOCs to find the host to feed or lay eggs, avoiding unsuitable plants [8–10]. The importance of the role of such secondary plant substances to determine the patterns of host plant utilization was first speculated by Fraenkel [11] in the famous article "The raison d'etre of secondary plant substances." On the other hand, the qualitative and quantitative modulation of VOC emission is also involved in the plant defense mechanisms, allowing it to affect the food searching of insects [12], deterring the herbivore (direct defense) [13,14], through antixenosis mechanisms [15,16], or calling its natural enemies (indirect defense) [17–19].

In citrus, the defensive action of plant secondary metabolites toward biotic factors, both pathogens and phytophagous insects, has been described [20,21]. As is happening in other plant species [22,23], the secondary metabolites of *Citrus* spp. have particular importance in the response to aphids avoiding the viability and proliferation of devastating plant pathogens to guarantee the host's survival in the environment [24]. Hijaz and co-workers [25] revealed that specific VOC classes, such as sabinene, (*E*)-β-ocimene, and linalool, are involved in biotic stress responses, including the huanglongbing disease (HLB)-associated bacterium, caused by *Candidatus* Liberibacter asiaticus (CLa), also known as "citrus greening." However, the mechanism driving the relationships between *Citrus* spp. and pathogen biotic vectors has not yet been explored.

Among insect vectors of viruses, aphids had historically the strongest impact on agriculture worldwide because of their ability to be carriers of several diseases [26,27]. In citrus, aphids showed a major role in the diffusion of the citrus tristeza virus (CTV), an aphid-borne closterovirus (genus *Closterovirus*, family Closteroviridae), ranked as one of the most important citrus diseases, that drastically modified the course of the citrus industry [28]. CTV infection causes a decline syndrome of different citrus species propagated on sour orange (*C. aurantium* L. Osb.) rootstock such as sweet oranges (*Citrus sinensis* L.), mandarins (*C. reticulata* Blanco), grapefruits (*C. paradisi* Macf.), kumquats (*C. japonica* Thumb.), or limes (*C. aurantifolia* Christm.). Sour orange was formerly the most important rootstock in almost every citrus-growing region worldwide [29] and is still the prevalent rootstock in the Mediterranean basin. Scions grafted to sour orange produce good yields of high-quality fruit that remain on the tree longer than most other common citrus rootstocks [30]. Sour orange plants are adaptable to a different range of soils [29], showing a considerable resistance to both abiotic factors, like frosts and salinity, and biotic stresses, such as gummosis and foot rot caused by *Phytophthora* spp. and the viroids citrus exocortis viroid (CEVd) and hop stunt viroid (HSVd). However, the high susceptibility of sour orange to CTV became a limit for their use as rootstock [31,32]. The strong impact of CTV epidemics, which caused the death of almost 100 million trees propagated on sour orange, generated the need for CTV-tolerant rootstocks that can resist well the major biotic and abiotic constraints in CTV-affected countries [33,34]. Indeed, due to efficient CTV dispersal by vectors, the propagation of citrus on CTV-tolerant rootstocks is the only viable option to manage CTV without tristeza decline [30,33].

In this context, among the CTV-tolerant varieties used as rootstock candidates to substitute sour orange, only few genotypes showed promising potential. Among these, Volkamer lemon (*C. volkameriana* Ten. & Pasq.) is a CTV-tolerant rootstock [35] with fast growth, highly suitable for lemon but not for orange, due to its poor quality in fruit production [36,37]. Another CTV-tolerant variety is Carrizo citrange [*C. sinensis* x *Poncirus trifoliata* (L.) Raf.]; however, it exhibits poor performance in clay, wet, and/or highly calcareous soils (pH ≥ 8.5), and it is susceptible to iron [38,39]. Finally, one of the most promising recently constituted CTV-tolerant citrus rootstock is Forner-Alcaide no. 5 (*C. reshni* Hort. Ex Tan. × *P. trifoliata*). This hybrid, whose use in new citrus orchards is constantly growing, shows resistance to both abiotic factors, such as salinity or lime-induced chlorosis, and biotic factors, such as the fungi *Phytophthora* spp. and the nematode *Tylenchulus semipenetrans* Cobb [40,41].

As the tolerance mechanism of these varieties toward CTV is not known, the VOC profile being a fast and noninvasive measure of phenotypic dynamics, allowing the association of plant phenotypes in accordance to plant disease resistance and/or stress tolerance [42], the study of constitutive plant secondary metabolites' profiles of CTV-susceptible and -tolerant citrus rootstocks could provide important information about the potential role of key metabolites to determine the virus tolerance through repellency toward its vector or directly affecting the pathogen. In fact, the plant VOC emitted in the air layer close to the plant may act to deter or attract aphids [43,44]. Specifically, *Aphis gossypii* Glover, one of the main CTV vectors, has been reported to show differential preference in host plant finding/acceptance in dependence of the host VOC's profile [45]. Furthermore, it is likely that the secondary metabolites sequestered in the plant tissue may act as a deterrent directly to the CTV-inducing tolerance mechanism. Among the plant secondary metabolites, terpenes are the largest class of plant compounds, and either constitutive or induced volatile mono- and sesqui-terpenes provide primary chemical defenses against insects and diseases [46–48]. Terpenes occur in mixtures (e.g., essential oils and resin) that enhance the effect of individual chemicals on insects and pathogenic microorganisms because of the additive and synergistic combinations of phytochemicals [49]. In this context, the constitutive percentages of leaf terpenes sequestered in tissues showed variability across species, but also within the species, families, varieties, and clones, with wider ecological consequences [50]. Therefore, knowledge of the constitutive profiles of volatile terpenes sequestered in leaf tissues is important to clarify the putative chemical defense mechanism of Citrus plants against aphids and CTV disease.

The present study aimed to describe the secondary metabolite profiles in four citrus varieties showing different tolerant grades to CTV infection, providing basic information for further studies. As VOC emissions can affect search behavior and the selection of host citrus plants for aphids, while the constitutive VOCs sequestered in leaf tissues can defend against the attack of both insects and virus, both emitted and constitutive VOCs from a CTV-susceptible species, the sour orange *C. aurantium* [31], and three CTV-tolerant genotypes, the *C. volkameriana*, the Carrizo citrange, and the Forner-Alcaide no. 5 [40,41], were compared.

## 2. Materials and Methods

### 2.1. Plants

Two-year old plants of *C. aurantium*, *C. volkameriana*, Carrizo citrange, and Forner-Alcaide no. 5 were provided from Vivai Maimone Giuseppe Alessio located in Milazzo (Messina—Italy). Citrus plants were produced and maintained in a greenhouse at $25 \pm 5 \,°C$ and 50–70% RH under a natural photoperiod. Four plants per variety were grown on a substrate consisting of sand and peat (1:1) in 20 L-cylindrical plastic containers and were watered 3 times per week and fertilized using alternating ratios of 3.1.1 and 1.3.1 (N.P.K). One month before starting the experiments, the plants were transferred to the IBBR Institute (Palermo—Italy) and were maintained in shadow recovery conditions until they were used for the collections and analysis of plant secondary metabolites.

### 2.2. VOCs Collection and Analysis

Chemical analysis of the plant secondary metabolites for each variety studied was carried out in parallel in two distinct laboratories. The VOC emission was analyzed through the headspace solid-phase micro-extraction (SPME) method [51] followed by gas chromatography and mass spectrometry (GC-MS) at the Department of Agricultural, Food and Forest Sciences (SAAF), University of Palermo. The leaves' metabolites analysis was performed at the Institute of Biosciences and Bioresources (IBBR), National Research Council of Italy (CNR), Via Madonna del Piano, Sesto Fiorentino (Florence—Italy), prior to placing the samples in hermetically sealed plastic bags and storing them at +4 °C until shipment in dry ice (with a 48 h courier).

### 2.2.1. VOCs Emission

VOCs emitted from *C. aurantium*, *C. volkameriana*, Carrizo citrange (*C. sinensis* × *P. trifoliata*), and Forner-Alcaide no. 5 were separately collected in the headspace by using SPME. The stationary phase used as a coating was polydimethylsiloxane (PDMS, 100 μm) (Supelco, Bellefonte, PA, USA). A manual SPME holder from the same manufacturer was used for injections. Fibers were conditioned in a gas chromatograph injector port, as recommended by the manufacturer, at 250 °C for 30 min.

For the headspace collections, four one-year old leaves chosen randomly from the four axes at the base of the petiole, from each plant belonging to the different varieties investigated, were cut and immediately covered with parafilm to minimize the VOC emission from the point of cutting. Leaves were then directly placed using forceps into 22 mL glass vials, which were sealed with a polytetrafluoroethylene silicon septum-lined cap (Supelco, Bellefonte, PA, USA). Subsequently, an SPME needle was then inserted through the septum and volatiles were absorbed on the exposed fiber for 5 min at a controlled room temperature (22 ± 1 °C). Therefore, the experiments were carried out using four biological replicates by sampling leaves each from individual trees belonging to each species, for a total of 16 samples (4 plants × 4 species).

In order to perform the chemical analysis of the collected VOCs, the loaded fiber was desorbed in the gas chromatograph inlet port for 2 min immediately after the end of the sampling time. Coupled GC-MS analyses of the headspace collections from the four plant species were performed on an Agilent 6890 GC system interfaced with an MS5973 quadruple mass spectrometer equipped with a DB5-MS column in splitless mode. Injector and detector temperatures were 260 °C and 280 °C, respectively. Helium was used as the carrier gas. The GC oven temperature was set at 40 °C for 5 min, and it was then increased by 10 °C/min to 250 °C. Electron impact ionization spectra were obtained at 70 eV, recording mass spectra from 40 to 550 amu. For each sample analyzed, VOC percentages were calculated by dividing the normalized peak area.

### 2.2.2. VOCs Sequestered in Leaf Tissues

Approximately 0.2 g of fresh weight per sample was placed in a glass vial and extracted with 1 mL of heptane. Each vial was sealed with a Teflon septum and crimped with an aluminum cap and then vortex-mixed for five minutes, sonicated for 15 min, and kept on overnight rotary agitation. After centrifugation at 4000 rpm for 10 min, the heptane phase was collected for the GC-MS analysis. An Agilent 7820 Gas Chromatograph system equipped with a 5977E MSD with EI ionization was employed, all from Agilent Tech. (Palo Alto, AC, USA). One microliter of extract in solvent was injected in a split/splitless injector operating in splitless mode. A Gerstel MPS2 XL autosampler equipped with the liquid option was used. The chromatographic settings were as follows: injector set at 260 °C, HP-innovax column (50 m, 0.2 mm i.d., 0.4 μm df); oven temperature program: initial temperature of 40 °C for 1 min, 5 °C min$^{-1}$ until 200 °C, 10 °C min$^{-1}$ until 220 °C, and then 30 °C min$^{-1}$ until 260 °C, with hold time of 3 min. The mass spectrometer was operating with an electron ionization of 70 eV, in scan mode in the *m/z* range of 29–330, at three scans per second. Data were acquired and analyzed using Agilent MassHunter software.

### 2.3. Statistical Analysis

Principal component analysis (PCA) of VOC profiles isolated from each variety was carried out using the R package FactoMiner [52] and factoextra (https://CRAN.R-project.org/package=factoextra, accessed on 27 January 2021). PCA enabled us to clearly assess what variables were able to discriminate the four samples investigated. A VENN diagram was also used to easily highlight the number of metabolites shared across varieties. The diagram was developed using the R package VennDiagram (https://cran.r-project.org/package=VennDiagram, accessed on 27 January 2021).

Relative contents (percentages or proportions) of volatile terpenes (VTs profiles) were calculated and compared among samples. Normality was not achieved after an arcsine square-root transformation, so tests were performed using the nonparametric Kruskal–Wallis rank-sum test followed by the Mann–Whitney U Test for multiple comparisons. Only significant differences ($p > 0.05$) were considered. Statistical analyses were performed using SYSTAT 12.0 (Systat Software Inc., USA) and GraphPad Prism 7.0 (GraphPad Software, Inc., La Jolla, CA, USA) software. Data are presented as percentage mean ± SE. A heatmap using the metabolites' profiles for each variety obtained through the solvent extraction approach was developed by heatmap.2 in R package gplots (https://github.com/talgalili/gplots, accessed on 29 January 2021).

To further validate the results observed, a Pearson correlation analysis between HS-SPME and the solvent extraction approach for each variety was also performed.

## 3. Results

### 3.1. VOCs Profiles Emitted

Thirty-six VOCs released from the Citrus varieties investigated via headspace SPME were recorded. The detected VOCs were distributed as follows: 11 monoterpene hydrocarbons, 4 monoterpene alcohols, 3 monoterpene aldehydes, 3 monoterpene esters, 13 sesquiterpene hydrocarbons, one sesquiterpene ester, and one aldehyde (Table 1). Monoterpenes were more abundant in the VOC profile of *C. aurantium* and *C. volkameriana*, with 92.47% and 92.31% of the total peak area, respectively; the remainder was mainly composed by sesquiterpenes. In contrast, VOC profiles of Carrizo citrange and Forner-Alcaide no. 5 were mostly characterized by sesquiterpenes, with nearly 78% of the total peak area, and by monoterpenes (22%). The detailed list of metabolites emitted from each sample is reported in Table 1.

**Table 1.** Mean percentage (±SE) of the volatile organic compound (VOC) emission from *C. aurantium* (CA), *C. volkameriana* (CV), Carrizo citrange (CC), and Forner-Alcaide no. 5 (FO) collected via headspace SPME.

| Peak | RT | LRI | Compound | Group | FO | CA | CV | CC |
|---|---|---|---|---|---|---|---|---|
| 1 | 9.357 | 925 | α-pinene * | Mt. hd. | 0 | 0.11 ± 0.03 | 0.92 ± 0.16 | 0 |
| 2 | 10.311 | 968 | sabinene | Mt. hd. | 0 | 0.22 ± 0.09 | 19.99 ± 1.94 | 0 |
| 3 | 10.416 | 973 | β-pinene | Mt. hd. | 0 | 2.57 ± 1.03 | 0 | 0 |
| 4 | 10.681 | 985 | myrcene * | Mt. hd. | 1.97 ± 2.38 | 4.46 ± 2.38 | 2.93 ± 0.88 | 2.52 ± 0.91 |
| 5 | 11.073 | 1003 | δ-3-carene | Mt. hd. | 2.13 ± 0.33 | 1.68 ± 0.33 | 0 | 4.18 ± 0.75 |
| 6 | 11.499 | 1026 | limonene * | Mt. hd. | 16.42 ± 7.07 | 2.83 ± 0.97 | 49.21 ± 3.51 | 40.01 ± 5.88 |
| 7 | 11.613 | 1032 | *cis*-β-ocimene | Mt. hd. | 0 | 1.25 ± 0.09 | 0 | 0 |
| 8 | 11.824 | 1044 | *trans*-β-ocimene | Mt. hd. | 1.12 ± 0.60 | 3.82 ± 0.23 | 2.76 ± 0.24 | 1.79 ± 0.54 |
| 9 | 12.065 | 1057 | γ-terpinene | Mt. hd. | 0 | 0.02 ± 0.00 | 0.25 ± 0.08 | 0 |
| 10 | 12.291 | 1069 | sabinene hydrate | Mt. est. | 0 | 0.06 ± 0.01 | 0.50 ± 0.19 | 0 |
| 11 | 12.562 | 1084 | α-terpinolene * | Mt. hd. | 0.14 ± 0.16 | 0.33 ± 0.02 | 0 | 0.38 ± 0.55 |
| 12 | 12.876 | 1101 | linalool * | Mt. alc. | 0 | 30.16 ± 3.05 | 6.60 ± 1.77 | 0 |
| 13 | 13.281 | 1127 | *allo*-ocimene | Mt. hd. | 0 | 0.69 ± 0.12 | 0.27 ± 0.09 | 0 |
| 14 | 13.681 | 1151 | citronellal | Mt. ald. | 0.60 ± 0.93 | 0.18 ± 0.02 | 6.06 ± 1.06 | 0.94 v 0.88 |
| 15 | 14.124 | 1178 | terpin 4-ol | Mt. alc. | 0 | 0.30 ± 0.04 | 0.13 v 0.05 | 0 |
| 16 | 14.421 | 1197 | α-terpineol * | Mt. alc. | 0.04 ± 0.07 | 0.11 ± 0.01 | 0.53 ± 0.22 | 0 |
| 17 | 14.821 | 1224 | citronellol * | Mt. alc. | 0.08 ± 0.15 | 0.13 ± 0.05 | 0.13 ± 0.08 | 0.02 ± 0.02 |
| 18 | 15.047 | 1239 | neral | Mt. ald. | 0 | 2.33 ± 0.34 | 0.72 ± 0.41 | 0 |
| 19 | 15.225 | 1251 | linalyl acetate | Mt. est. | 0 | 38.05 ± 5.20 | 0 | 0 |
| 20 | 15.487 | 1268 | citral * | Mt. ald. | 0.12 ± 0.23 | 3.14 ± 0.64 | 0.53 ± 0.33 | 0 |
| 21 | 16.035 | 1306 | undecanal | Aldehyde | 0 | 0 | 0.05 ± 0.03 | 0 |
| 22 | 16.458 | 1336 | bicycloelemene | Sqt. hd. | 1.36 ± 0.43 | 0.16 ± 0.04 | 0.55 ± 0.21 | 0.64 ± 0.24 |
| 23 | 16.493 | 1339 | δ-elemene | Sqt. hd. | 1.99 ± 0.33 | 0.18 0.04 | 0.88 ± 0.27 | 0.99 ± 0.23 |
| 24 | 16.716 | 1355 | neryl acetate | Sqt. est. | 0 | 1.11 ± 0.37 | 0.11 ± 0.06 | 0 |
| 25 | 16.989 | 1375 | geranyl acetate | Mt. est. | 1.88 ± 1.57 | 2.63 ± 0.50 | 0.15 ± 0.14 | 0.71 ± 0.07 |
| 26 | 17.080 | 1382 | α-copaene | Sqt. hd. | 0.22 ± 0.14 | 1.25 ± 0.52 | 0 | 0.07 ± 0.06 |
| 27 | 17.241 | 1393 | β-elemene | Sqt. hd. | 8.37 ± 1.36 | 0 | 0 | 4.47 ± 0.83 |
| 28 | 17.716 | 1429 | *trans*-β-caryophyllene * | Sqt. hd. | 30.40 ± 4.73 | 1.40 ± 0.50 | 2.96 ± 0.84 | 23.65 ± 2.69 |
| 29 | 17.764 | 1433 | γ-elemene | Sqt. hd. | 6.55 ± 0.97 | 0.04 ± 0.01 | 0.23 ± 0.19 | 2.48 ± 1.07 |
| 30 | 17.802 | 1436 | *trans*-α-bergamotene | Sqt. hd. | 1.95 ± 0.189 | 0 | 0.88 ± 0.24 | 1.31 ± 0.21 |
| 31 | 17.988 | 1450 | *cis*-β-farnesene * | Sqt. hd. | 9.16 ± 0.93 | 0.29 ± 0.11 | 0.04 v 0.04 | 7.49 ± 1.69 |

**Table 1.** *Cont.*

| Peak | RT | LRI | Compound | Group | FO | CA | CV | CC |
|------|------|------|----------|-------|------|------|------|------|
| 32 | 18.177 | 1466 | humulene | Sqt. hd. | 2.42 ± 0.61 | 0.12 ± 0.02 | 0.17 v 0.10 | 2.89 ± 1.39 |
| 33 | 18.490 | 1490 | germacrene D | Sqt. hd. | 10.17 ± 2.56 | 0.11 ± 0.02 | 0.37 v 0.11 | 4.35 ± 1.34 |
| 34 | 18.672 | 1504 | bicyclogermacrene | Sqt. hd. | 2.02 ± 0.66 | 0.23 ± 0.03 | 0.63 ± 0.11 | 0.88 ± 0.23 |
| 35 | 18.736 | 1509 | β-bisabolene * | Sqt. hd. | 0.76 ± 0.25 | 0 | 0.61 v 0.16 | 0.19 ± 0.10 |
| 36 | 18.903 | 1523 | δ-cadinene | Sqt. hd. | 0.12 ± 0.08 | 0.01 ± 0.00 | 0.06 ± 0.00 | 0.03 ± 0.01 |

RT, retention time; LRI, linear retention index calculated using an n-alkane series (C9–C31) in hexane, injected under the same conditions as samples. * Compound confirmed by matching retention time and mass spectra with authentic standards. Groups: Mt., monoterpenes (C10); Sqt., sesquiterpenes (C15); subgroups: hd. hydrocarbons; ald., aldehydes; alc., alcohols; est., ester.

The VOC emission recorded underlined qualitative and quantitative differences across the different Citrus varieties. In the PCA analysis, the two first components contributed around 90% to the overall variability among the four varieties studied, displaying a clear separation among them (Figure 1). The compounds more positively correlated with the first component (PC1) were mainly linked to monoterpenes (green), while sesquiterpenes (blue) were mainly negatively correlated in the same component. Nearly sixty-five percent (65%) of metabolites positively linked to PC2 belonged to monoterpene and, in the same quadrant, undecanal, the only aldehyde isolated, was also found. A clear discrimination across genotypes according to both their leaf morphology and susceptibility to CTV infection was shown (Figure 1). The CTV-tolerant samples *C. volkameriana* L., and the pair Carrizo and Forner-Alcaide no. 5, the two genotypes obtained from trifoliata parents, were in the up-right and left quadrants, respectively, while the CTV-susceptible *C. aurantium* L. was in the lower right quadrant, separated from the others. Nearly forty-two percent (42%) of metabolites (15) isolated were shared across varieties (Table 1; Figure 2). Interestingly, two compounds (trans-α-bergamotene and β-bisabolene) were recorded in all CTV-tolerant varieties; on the contrary, β-pinene, cis-β-ocimene, and linalyl acetate were extracted only from the susceptible *C. aurantium* (Table 1; Figure 2). Additionally, linalyl acetate, trans-*β*-caryophyllene, and limonene were the most abundant metabolites in *C. aurantium*, Forner-Alcaide no. 5, and the pair *C. volkameriana*/Carrizo citrange, respectively. Finally, β-elemene was found only in the trifoliate genotypes (Carrizo citrange—4.47%, and Forner-Alcaide no. 5—8.37%).

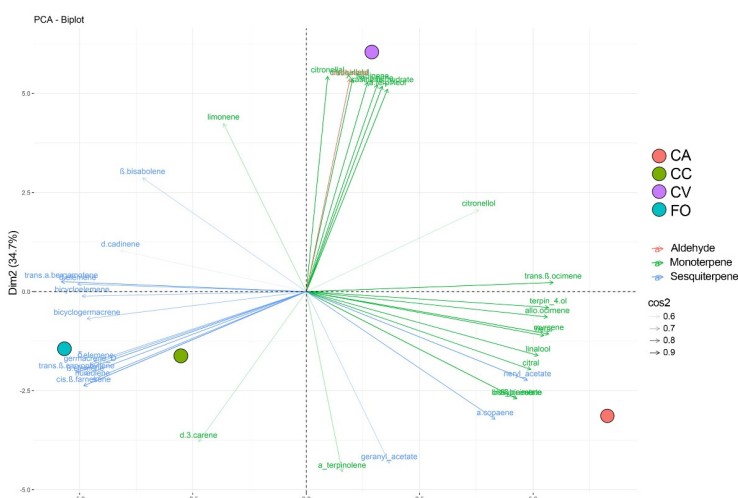

**Figure 1.** Biplot of the principal component analysis (PCA) for VOC profiles detected on *Citrus* varieties showing different susceptibilities to CTV infection. Based on their profiles, samples were organized in four groups, and the associated compounds to varieties separation are indicated by vectors in the plot, underlining their significance values (0.6 < cos2 < 0.9). Each main category is highlighted with a different color: aldehyde (red), monoterpene (green), and sesquiterpene (blue). CV: *Citrus volkameriana*; FO: Forner-Alcaide no. 5; CC: Carrizo citrange; CA: *Citrus aurantium*.

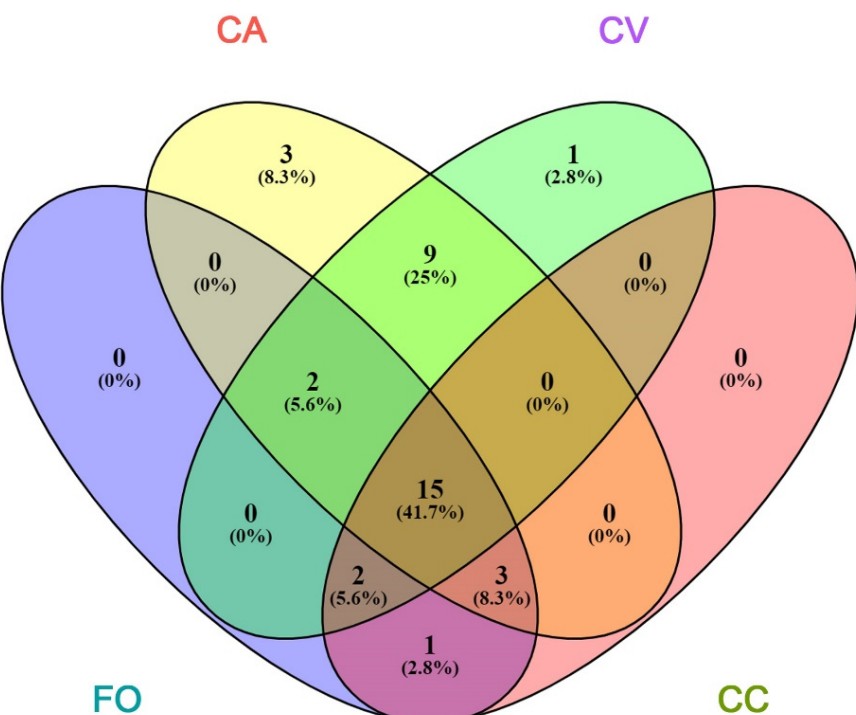

**Figure 2.** Venn diagram of the 36 mass peaks of the volatile organic compound (VOC) emitted from the four varieties investigated. CV: *Citrus volkameriana*; FO: Forner-Alcaide no. 5; CC: Carrizo citrange; CA: *Citrus aurantium*.

### 3.2. VOCs Sequestered in Leaf Tissues

The GC-MS analysis of leaves of the four *Citrus* varieties detected 31 metabolites. In particular, the constitutive leaf VOCs were composed of 12 monoterpene hydrocarbons, 5 monoterpene alcohols, 2 monoterpene aldehydes, 2 monoterpene esters, one monoterpene alkylbenzene, 8 sesquiterpene hydrocarbons, and a sesquiterpene not identified. The Kruskal–Wallis ANOVA test showed significant differences between different citrus varieties (d.f. = 3, *n* = 16) in the relative contents of all the VOCs, except camphene, α-terpineol, geranyl acetate, β-citronellol, elixene, and sesquiterpene (Figure 3; Supplementary Table S1).

Overall, the *C. aurantium* variety showed significantly lower values of several metabolites, such as α-pinene, α-terpinene, ocimene, p-cymene, δ-elemene, and γ-elemene, compared the other samples, but it was characterized by a significantly higher relative content of both linalyl acetate and linalool than tolerant genotypes. On the contrary, the three tolerant varieties were characterized by considerable contents of limonene and citronellal, significantly higher than *C. aurantium*. Among tolerant genotypes, *C. volkameriana* showed a significantly higher proportion of citronellal in comparison to Carrizo citrange and Forner-Alcaide no. 5 (Figure 3; Supplementary Table S1), and had the greatest content of sabinene, significantly higher than those of other samples. The profile belonging to *C. volkameriana* was also characterized by significantly higher values of nonanal and a lower amount of elixene compared to the other varieties. In addition, the proportions of myrcene, β-caryophyllene, β-ylangene, and germacrene were significantly lower in this genotype than in Carrizo citrange and Forner-Alcaide no. 5. The β-pinene and nerol profiles were shared by *C. volkameriana* and *C. aurantium*, and their amounts were significantly higher than that of Carrizo citrange; Forner-Alcaide no. 5. *Citrus volkameriana* and *C. aurantium* profiles also showed small contrast profiles in the camphene and α-terpineol contents, respectively, significantly higher and lower in *C. aurantium* than *C. volkameriana* (Figure 3; Supplementary Table S1). Finally, the Carrizo citrange profile showed a significantly higher relative content of α-humulene compared to both *C. aurantium* and *C. volkemariana*,

while the amounts of 3-carene, α-phellandrene, γ-terpinene, terpinolene, geraniol, and β-farnesene were significantly lower in C. *volkameriana* than in the other varieties.

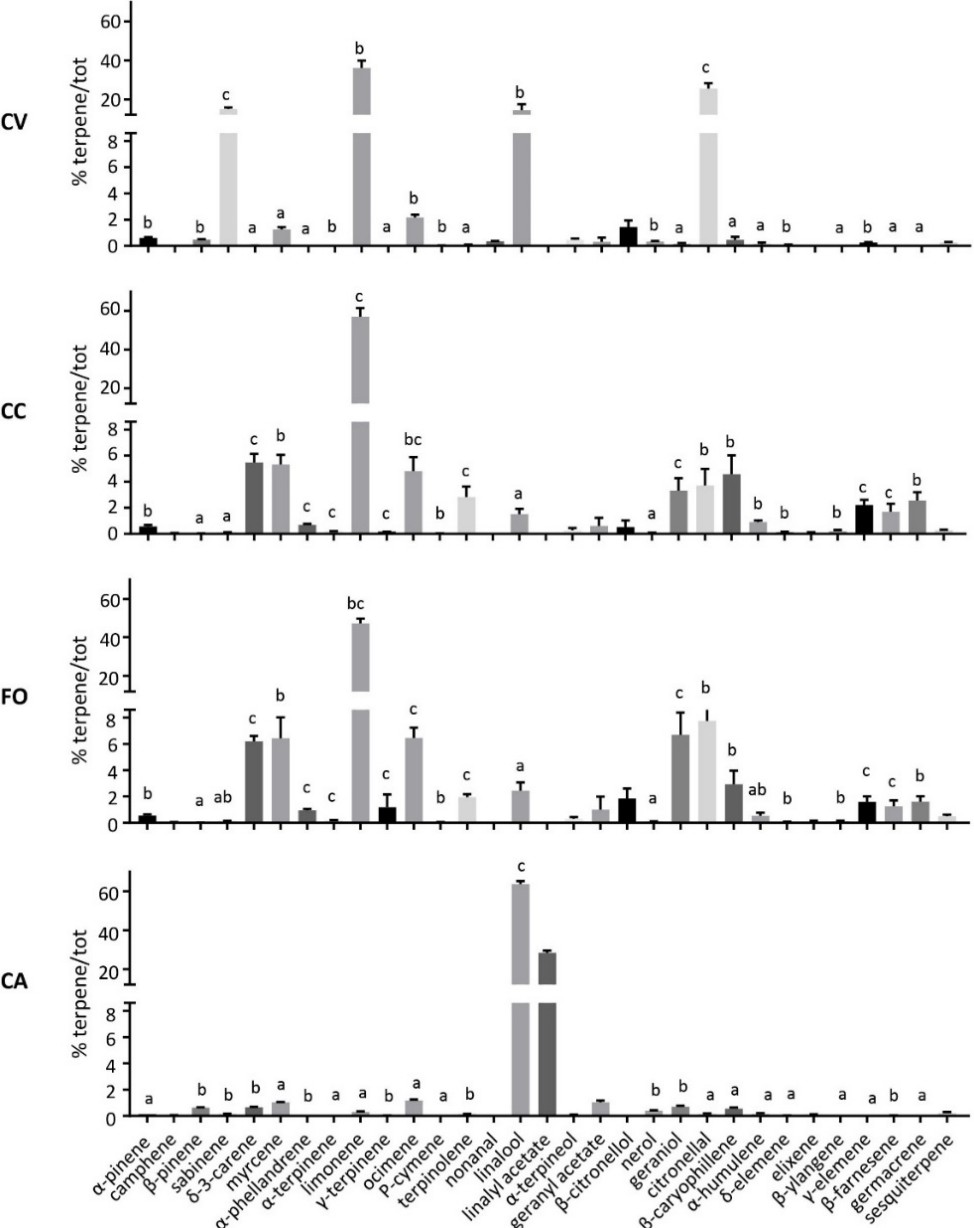

**Figure 3.** Percentages of the plant secondary metabolites sequestered in the leaves of Citrus volkameriana (CV), Forner-Alcaide no. 5 (FO), Carrizo citrange (CC), and *Citrus aurantium* (CA). Bars indicate means + standard errors. Different letters on the top of the bars indicate significant differences between the four Citrus varieties (Mann–Whitney U test).

A heatmap based on metabolite profiles isolated through the solvent extraction approach clustered the four varieties in two main distinct groups (Figure 4), allowing one to easily highlight the differences across genotypes. The first cluster enclosed only the CTV-susceptible *C. aurantium*, while in the second, one of the CTV tolerance varieties was grouped (Figure 4). In this last cluster, Forner-Alcaide no. 5 and Carrizo citrange, the two varieties obtained from trifoliate genotypes were in the same branch.

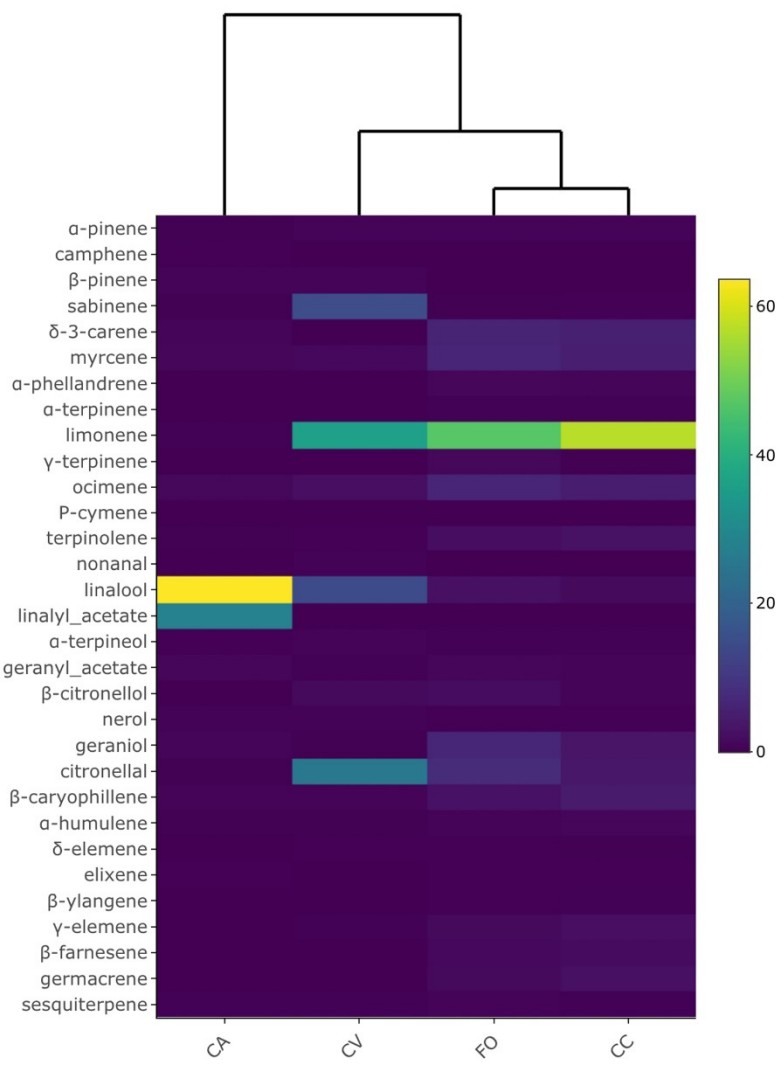

**Figure 4.** Heatmap on plant secondary metabolites sequestered in the leaves of *Citrus volkameriana* (CV), Forner-Alcaide no. 5 (FO), Carrizo citrange (CC), and *Citrus aurantium* (CA). The heatmap legend (right) indicates the abundance (low—blue; high—yellow) of metabolites recorded in each sample.

The results obtained by GC-MS were in agreement with the data recorded through headspace SPME, with nearly 68% (21 metabolites) shared. Indeed, the correlation analysis carried out to verify the data reliability and reproducibility obtained from the two approaches underlined a strong coefficient index ($R = 0.79$), with high significance ($p < 0.001$) (Supplementary Figure S1).

## 4. Discussion

Citrus leaf volatiles with different tolerance behaviors to CTV were analyzed by using both SPME or heptane extraction followed by GC-MS. The results recorded indicated that the VOC emitted/sequestered in the leaves belonged mainly to volatile terpenes (VTs), showing strong differences among the varieties investigated.

In detail, the secondary metabolites' profiles emitted from *C. aurantium* and *C. volkameriana* were characterized by monoterpenes, representing more than 90% of their total emission, while the Forner-Alcaide no. 5 and Carrizo citrange profiles highlighted a higher amount of sesquiterpenes. The role of sesquiterpenes as deterrents versus piercing sucking insects, vectors of important plant diseases, has been widely reported [53,54]. Therefore, the VOCs profiles of Carrizo citrange and Forner-Alcaide no. 5 could be related to CTV tolerance through the discourage of its aphid vectors. Indeed, β-caryophyllene, the main

sesquiterpene emitted by Forner-Alcaide no. 5 and Carrizo citrange, has been reported to have both a repellent effect on the psillid *Diaphorina citri* Kuwayama [55], a vector of the yellow shoot disease, and a toxic effect on *A. gossypii*, one of the main vectors of CTV in Europe [56]. In agreement, germacrene D, another sesquiterpene recorded in the same genotypes, has also been shown to repel the grain aphid *Sitobion avenae* F. (Hemiptera; Aphidoidea) [57]. Recently, Wang and collaborators [58] highlighted that higher amounts of sesquiterpenes, such as β-caryophyllene, humulene, α-bergamotene, and β-bergamotene, were repellent toward the *Macrosiphum euphorbiae* L. (Hemiptera; Aphidoidea), also inducing a reduced longevity and fecundity of aphids. Finally, the plant sesquiterpenes emission can be activated by the aphid feeding action [59,60], suggesting a possible role of these volatiles as a direct defense by deterring the aphid or indirectly by recruiting its natural enemies. Furthermore, high amounts of caryophyllene, γ-elemene, β-elemene, and germacrene D were found in *Severinia buxifolia* Poir., a tolerant cultivar to *Candidatus liberibacter asiaticus* (CLa) belonging to the Rutaceae family [61]. Moreover, capsidiol 3-acetate, a sesquiterpenoid phytoalexin, was reported as a basal defense antiviral compound produced against Potato Virus X (PVX) in *Nicotiana benthamiana* Domin. [62]. Finally, β-caryophyllene and germacrene D, characterizing more than 40% of the total VOC recorded in the Forner-Alcaide no. 5 genotype, were the main components of *Teucrium* spp essential oils, and they have been found to be effective anti-phytovirus compounds against the cucumber mosaic virus (CMV) [63].

Interestingly, trans-α-bergamotene and β-bisabolene were emitted only by the three tolerant genotypes (Table 1, Figure 2). Both metabolites were involved in the mechanisms of plant disease resistance, modulating the plant–pathogen interactions though Jasmonic acid (JA) signaling, an important hormone involved in the defense system [64–66]. A high induced expression of terpene synthase (TPS) genes (like *JAmyb*) associated with strong sesquiterpenes production, including β-bisabolene, was found in a resistant line of rice to green leafhopper [66]. In maize, the pathogen-induced sesquiterpenoid accumulation was directly associated with high transcript levels of *Tps6* and *Tps11* [67], and mutant plants for these genes exhibited increased susceptibility to *Ustilago maydis* Corda [68]. Similarly, the trans-α-bergamotene emission elicited by herbivories exhibited in *Nicotiana* spp. acts as an indirect defense by attracting predators of *Manduca sexta* L. (Lepidoptera; Sphingidae) larvae and eggs in the *N. attenuata* Torr. native habitat [64].

In agreement, the volatile metabolite profiles sequestered in the leaf tissues of the four Citrus varieties evidenced a strong difference in the relative amounts of several VTs that might have important ecological roles [69]. Indeed, the constitutive VOC sequestered in the leaves of the most CTV-susceptible variety, *C. aurantium*, showed a significantly low level of specific VTs such as limonene, α-pinene, and p-cymene that are known to be largely present in plant extracts with deterrent effects toward several insect species at adult or egg stage [70–75]. These chemicals have strong influence on the ecology of virus carriers such as aphids, by directly deterring them or indirectly by recruiting their natural enemies. Among VTs differentially recorded between tolerant and susceptible genotypes, ocimene and limonene have been reported as aphid-deterrent compounds [76], while α-pinene was demonstrated to be an attractant to aphid natural enemies such as *Altica cyanea* (Weber) (Coleoptera: Chrysomelidae) and *Lectocybe invasa* Fisher (Hymenoptera: Eulophidae) [77,78]. All these terpenes were more abundant in Forner-Alcaide no. 5, *C. volkameriana*, and Carrizo citrange rather than in *C. aurantium.* The citronellal's amount, a compound previously reported as an inhibitor of plant infection from the tobacco mosaic virus [79], was also noteworthy and higher in the tolerant varieties than *C. aurantium*. On the contrary, linalool, a VT highly expressed in C. *aurantium*, and already observed in citrus essential oils profiles [80,81], showed a lower insecticidal activity [82]. The biocidal properties of some terpenes may be explained by their interaction with the insect octopamine receptor. Octopamine is a neurotransmitter synthesized in insects that can modify their behaviors, metabolism, and locomotion [83]. p-Cymene and pinene are terpenes with strong octopaminergic receptor antagonist activity that could explain their

high insecticidal activity [71], while the lower activity of linalool could be related to the lack of octopamine receptor interaction [82].

The accumulation of several metabolites, including limonene, was previously observed in Citrus after CLa infection [84], highlighting their possible role as antimicrobial compounds against infection. Numerous natural compounds are produced in plants in response to biotic stresses, through the activation and/or the increase in several pathways implicated in plant defense [85–87]. Metabolomic and transcriptomic studies underlined an induced citrus plant defense response, with evident changes in both primary and secondary metabolites [86,87], such as aromatic amino acids used as precursors for many compounds implicated in biotic stress responses [84,88].

Our current results, together with previous evidence [86,87], highlight a possible relationship between constitutive citrus' VTs and the behavior against CTV infection, through a direct or indirect system. The VOC-emitted profiles recorded in the different Citrus varieties shed light on a possible role of VTs in the chemical defense mechanisms of plants to pest and diseases, as previously reported in other species [89–91]. Induced terpenes also have a key role in the defense mechanism against insects and pathogens in plants [89,90,92]. Significant differences in the constitutive monoterpenes sequestered in cortical tissues between two clones of Norway spruce selected for resistance to root and butt rot disease by the fungus *Heterobasidion parviporum* Niemelä & Korhonen were found [93]. In particular, a higher relative content of constitutive δ-3-carene was shown in the resistant compared to the susceptible plants. Besides, *H. parviporum* infection caused an increase in δ-3-carene proportions in both clones, suggesting its potential defensive role against the attack of this pathogenic fungus. Upon wounding, constitutive toxic terpenes are released against the insects, inhibiting the pathogen growth and performing a barrier for further infections [90]. An over-expression of β-ocimene from *N. tabacum* L. triggered a high discharge of methyl salicylate and cis-3-hexen-1-ol in *Lycopersicum esculentum* Mill., both of which are proposed to cause deterrence on aphid development and reproduction, increasing the defense ability of tomato against these pests [94]. Although the plant volatile–herbivore interactions have been widely studied, the knowledge of volatile–microbe (phytopathogenic fungi, bacteria, and viruses) interactions is in embryos and is poorly documented [89]. The information on the role of VTs in the plant resistance/antixenosis to aphids and their related diseases might allow the selection of tolerant or resistant plants by traditional and unconventional plant breeding, reducing several diseases in both agriculture and forestry.

Considering the importance that the varieties studied in this work have as rootstock for citrus industry, especially after the spread of CTV epidemics in Mediterranean countries, the information presented here is the starting point for future studies. In fact, although the Citrus rootstocks with different behaviors to CTV infection showed dissimilar VT profiles, the role of specific terpenes in the pathogen tolerance and the mechanisms driving a possible induction of functional VOCs in the scions, as reported in other species [95,96] and in citrus for CLa disease [97], are of primary importance and must still be explored. To fill this gap, our next studies will investigate through next-generation sequencing approaches if the four rootstocks evaluated here can influence the VOC emission of grafted *C. sinensis*. Furthermore, our efforts will also focus on behavioral experiments, testing different aphid species in the laboratory to evaluate their preferential response to the *C. aurantium*, *C. volkameriana*, Carrizo citrange, and Forner-Alcaide no. 5.

## 5. Conclusions

The results obtained in this study evidence the strong differences in terms of constitutive VTs emitted or sequestered in the leaves of four citrus varieties characterized by different tolerance grades to the CTV. This can suggest that the CTV-tolerance mechanism elicited by *C. volkameriana*, Carrizo citrange, and Forner-Alcaide no. 5 may be mediated by the VTs, through inhibition of the virus itself or deterrence toward its insect vectors.

**Supplementary Materials:** The following are available online at https://www.mdpi.com/article/10.3390/agronomy11061120/s1, Figure S1: Correlation analysis between HS-SPME and solvent

extraction, Table S1: Relative contents of terpenes (means ± standard error) of Citrus volkameriana (CV), Carrizo citrange (CC), Forner-Alcaide no. 5 (FO) and Citrus aurantium (CA).

**Author Contributions:** Conceptualization, S.G., L.A., F.M., S.F.D.B. and M.M.; methodology, S.G., M.M.; formal analysis, S.G., M.A.A., M.M., F.M.; investigation S.G., M.A.A., M.M., G.C., E.P., L.A., A.M.; data curation, F.M., S.G., M.M.; writing—original draft preparation, S.G., M.M., E.P.; All the author have visualized and reviewed the manuscript. All authors have read and agreed to the published version of the manuscript.

**Funding:** This research received no external funding.

**Data Availability Statement:** The data presented in this study are available in the article or supplementary material.

**Acknowledgments:** The authors are grateful to Stefano Colazza, Head of the Department of Agricultural, Food and Forest Sciences of the University of Palermo, for giving us hospitality in the Department's laboratories during this research, and to Vivai Maimone Giuseppe Alessio (http://www.maimonevivai.it) for the plant material supply.

**Conflicts of Interest:** The authors declare no conflict of interest.

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
