# Peer review of "Citrus Varieties with Different Tolerance Grades to Tristeza Virus Show Dissimilar Volatile Terpene Profiles"

_agronomy, doi:10.3390/agronomy11061120_

Round 1
Reviewer 1 Report
In this manuscript the authors present the analysis of secondary metabolites and more precisely of volatiles organic compounds (VOCs) in four citrus varieties with different susceptibility/tolerant phenotypes against Tristeza disease. Comparison of the profile of the emitted VOCs and the VOCs sequestered in leaf tissues showed a possible relationship between the tolerant phenotype and the qualitative/quantitative content in volatile terpenes, that could act directly against citrus tristeza virus or against its aphid vector. The identification and characterization of VOC profiles associated with resistance/tolerance against Tristeza disease will help identify novel rootstocks with similar characteristics.
Overall, the manuscript is well written and gives important and novel information about the resistant/tolerant phenotype against Tristeza disease in three widely-used citrus varieties. However, the total number of plants assessed per citrus variety in the VOCs profile analysis is not clear. The authors should clearly indicate if their replicated are derive from individual plants or are a composite sample. I recommend the publication of the manuscript with minor modifications and the addition of the above information in Materials and Methods section. My comments to the authors are the following:
Line 57: Change to “… of Citrus sp. a particular importance…”
Line 61: Change to “… huanglongbing (HLB) disease…”
Line 64: Change to “Among insect vectors of viruses…”
Line 65: Change to “… their ability to be carriers…”
Line 72: Change to “…citrus-growing region worldwide…”
Line 77: Change to “… and the viroids citrus exocortis viroid (CEVd) and hop stunt viroid (HSVd)”
Line 85: Change to “… with fast-grow and is considered highly…”
Line 87: Change to “…however it exhibits poor performance in clay…”
Line 91: Change to “…shows resistance to both abiotic factors…”
Line 98: Change to “In specific…”
Line 112: Change to “The present study aims…”
Line 122, line 152: The authors should state the number of individual plants assessed per variety in each analytical method. The authors should indicate if the four replicates per variety are derived from individual trees.
Line 214: Change to “…genotypes according to both their leaf….”
Line 278: Change to “…one the CTV tolerant varieties….”
Line 286: Change to “Figure 3. Percentages of the plant secondary….”
Line 294: An explanation of what different colours indicate (and the number close to the colour bar) in the heatmap should be added in the legend of the figure.
Line 306: Change to “…through the discourage of its aphid vectors.”
Line 320 and 324: According to ICTV recent instructions virus names should not be capitalized.
Line 333: Change to “…exhibited in Nicotiana sp.,…”
Line 341: Change to “These chemicals have strong influence on the ecology of virus carrier like aphids, by directly deterring them or indirectly by recruiting their natural enemies.”
Lines 359-360: These lines should be rephrased. The meaning of this sentence is not well understood.
Line 364: Change to “…with previous evidences [80,81], highlight…”
Line 366: Change to “…shed light to a possible…”
Line 373: Change to “…δ-3-carene was showed in the resistant comparing to the susceptible plants.”
Line 383: Change to “…diseases, might allow…”
Line 384: “ thus volatile plants to reduce disease in agriculture and forestry”. This part of the sentence should be rephrased.
Line 387: Change to “…presented here are the starting point for future studies.”
Line 390: Change to “…CLa disease [91] are of primary importance…”
Reviewer 2 Report
The present study is aimed to evaluate the relationships the spectrum of secondary metabolites and the resistance of four citrus varieties to CTV infection. This study determined the volatile metabolites in the leaves of four citrus varieties with different levels of resistance to CTV, and discussed the differences in metabolism patterns between varieties. Two of them have strong resistance to CTV. Citrus varieties (Carrizo citrange and Forner-23Alcaide no. 5) have higher sesquiterpene content in VOCs. Some of the three CTV-tolerant varieties have a higher content of monoterpenes. But the content of this study is simple, and there are not enough experiments to prove the possible ecological consequences of such differences in term of tolerance grade to CTV and/or its related vectors. In the manuscript, the interaction mode between volatile metabolites and citrus virus invasion vectors was proposed and discussed, but the author did not verify this in the manuscript. Additional data support and supplementary description are needed. This study has no outstanding technological breakthroughs and clear logical thinking. Additional examination and supplementary descriptions in details are needed in some cases:
- The author needs to further verify that in CTV-resistant varieties, monoterpenoids such as limonene and cymene have significant control effects on invading carriers. It is recommended to verify that these substances have control effects on pests.
- The author mentioned in the abstract the possible ecological consequences of differences in the tolerance levels of CTV and its related media, but this study have not the relevant details in the article, it is recommended to consult the relevant literature and supplement the description.
- The four charts CV, CC, FO, and CA in Figure 3 in the manuscript share the same abscissa. It is difficult to find relevant data when reading. It is recommended to add the corresponding substance name to the abscissa of each chart.
Round 2
Reviewer 2 Report
The authors have resolved my concerns.